# Using Social Media to Collect Dietary Information for Public Health Policy

**DOI:** 10.3390/nu14245322

**Published:** 2022-12-15

**Authors:** Catherine Rycroft, Sarah Beer, Nicola Corrigan, Janet Elizabeth Cade

**Affiliations:** 1Nutritional Epidemiology Group, School of Food Science and Nutrition, University of Leeds, Leeds LS2 9JT, UK; 2Dietary Assessment Ltd., Nexus, Discovery Way, University of Leeds, Leeds LS2 3AA, UK; 3Office for Health Improvement and Disparities, Department of Health and Social Care, Blenheim House, Leeds LS1 4PL, UK

**Keywords:** online survey, myfood24^®^, nutritional analysis software, public health, policy, social media, regional survey

## Abstract

There is no regular, routine measurement of food and nutrient intake regionally in the UK. Our goal was to identify a method to support policy makers tracking the local population food intakes. The aim of this study was to test the feasibility of using social media to obtain a large sample in a short time, with a regional focus; collecting dietary information using online tools. A Facebook (FB) boost approach was used to recruit a regional (Yorkshire and Humberside) sample of adults to complete a brief online survey followed by a detailed measure of food and nutrient intakes for the previous day using myfood24^®^. The FB posts were boosted for 21 days and reached 76.9 k individuals. 1428 participants completed the main questionnaire and 673 participants completed the diet diary. The majority of respondents were older women. 22% of respondents reported experiencing moderate food insecurity during 2021. Overall nutrient values recorded were similar to national survey data. Intakes of fibre and iron were low. Despite some challenges, this study has demonstrated the potential to use social media, in this case Facebook, to recruit a large sample in a short timeframe. Participants were able to use online tools to report food and nutrient intakes. This data is relevant to local and national policy makers to monitor and evaluate public health programmes.

## 1. Introduction

Local public health research which evaluates food strategies at individual intake level is limited due to lack of relevant data. Ideally, population-based surveys should recruit participants who are representative of the underlying target population. However, in traditional surveys recruitment costs can be high with low response rates, making considerable time and personnel demands [1]. Globally, over 4.5 billion people use the Internet and over 3.8 billion use social media [2]. Web-based recruitment through social media is promising; social media penetration in the UK is 90%, with 46.6 m Facebook users [3]. Facebook allows for targeted advertisements to preferentially reach people based on demographics, location, interests, and behaviours [4], so could be ideal for regional surveys. A recent literature review has shown that use of social media is a cost-effective and efficient strategy for recruiting research participants [5]. Nutritional analysis software with accurate underpinning food composition data focused on the UK could enable quick and easy large scale dietary data collection [6].

The UK National Food Strategy (June 2022) [7] identifies changes needed to the national diet by 2032, to meet health and climate commitments. Access to and consumption of healthy, low emission foods such as wholegrains, fruits and vegetables is generally low, especially in lower income/educational groups [8]. Food insecurity in UK households is a substantial and growing concern [9]. In May 2020, there were nearly 5 million people in the UK experiencing food insecurity, including 1.7 million children; this represents a 250% increase over levels seen before the COVID-19 outbreak [10]. 

Policy makers require understanding of population food intake, at the local and national level [11], along with health and sustainability footprints. For example, reducing the carbon footprint of meat intake, increasing fruits and vegetables and fibre and reducing high fat salt and sugar foods. Measurement of food and nutrient intakes can help evaluate local and national policies. This includes impacts of food product reformulation, for example reduction of sodium or trans-fatty acid intakes [12]. Measurement of food intake can provide a better understanding of how food insecurity is affecting individuals [13]. Ensuring a good baseline measure of food and nutrient intakes can also support development of future health policies through understanding which population groups meet national dietary guidelines [14]. This has the potential to highlight population groups in most need of support or intervention. Social media interventions can positively change diet-related behaviours. In addition, to measuring intakes, social media contacts can provide opportunity for intervention through gamification, image sharing and group chats [15].

The regional, Yorkshire and Humber Public Health Community of Improvement for Healthy Weight, Nutrition and Food is developing healthier and resilient food systems to support national and local obesity strategies and for equitable food access and provision [10]. How policy makers should best reach their local population to ensure wide representation of views on a limited budget is unclear. Some research has suggested that unpaid recruitment channels may be more effective than paid channels, but that these different approaches result in samples with different characteristics [16]. Social networking platforms offer a wide reach for public health communication and interventions allowing access to broad audiences using tools that are straightforward to use [2]. They also support a degree of tailoring to contact harder to reach groups. On the other hand, there is a risk that people who experience health inequity may face barriers to the use of social media through limited access, reading or electronic health literacy. Poorer e-health literacy has been associated with being older, having less education, and not having access to a mobile device [17]. A systematic review found that older adults were less likely to take part in research using social media recruitment [5]. In relation to research costs, use of social media boosts can be more cost effective than traditional methods of recruitment [5].

There is no regular, routine measurement of food and nutrient intake regionally in the UK. Our goal was to support policy makers tracking the local population food intakes. The aim of this study was to test the feasibility of using social media to obtain information on food and nutrient intakes in a large population sample. Our objectives were as follows: (1) to use Facebook to contact a regionally representative sample of adults; (2) to achieve the study recruitment in a short time frame; (3) to collect detailed dietary intake information using online tools.

## 2. Materials and Methods

We developed and undertook a rapid, time limited, regional (Yorkshire and Humberside) food and nutrient tracking survey for use with policy makers. Eligible respondents were asked for informed consent before completing a short online questionnaire (see Section 2.3) and reporting their food and drink intake for the previous day.

### 2.1. Using Facebook to Recruit Study Participants

We recruited adults through Facebook, using a ‘boost’ approach, during January and February 2022. This required a Facebook business account and profile page for the study. Targeted Facebook users, specifically adults living in the Yorkshire and Humber region, saw boosted (paid for) posts, inviting them to take part. A boosted post is a post to the study page’s timeline where money is paid to boost the post to a specific audience.

The study was called ‘One day: Diet in Yorkshire and Humber’, and a Facebook page was created to promote the study. The page included a profile picture with a study logo, a cover photograph, ‘About’ information for potential participants and a welcome message: 

‘Welcome to One Day: Diet in Yorkshire and Humber, finding out what adults living in the region eat and drink. If you are interested in joining our survey and seeing some feedback about your nutritional intake, please read the ‘About’ section or select the blue Sign up button. You can enter our Prize Draw for the chance to win a shopping voucher for £100, £50 or £10! We are recruiting participants until 20 February 2022.’ An example of a study FB page can be found in Appendix A.

A call-to-action ‘Sign up’ button for visitors to the page was added to the Facebook page header using an URL that linked to the survey’s main questionnaire in Online Surveys. The same URL was also added to the ‘About’ information and embedded in posts on the Facebook page.

During late January and February 2022 four Facebook posts, each with a photograph and the URL link to the questionnaire in Online Surveys, were written and posted on the Facebook page. Three of the four posts were ‘boosted’, so that they appeared in the timeline of a target audience of Facebook users. For each boost we specified the target audience, the duration of the boost (date range and/or number of days to promote the post) and a maximum total budget. A debit card was registered with the Facebook account. Facebook took payments at intervals as each boost progressed, not exceeding the total budget set for that boost. 

We wanted to recruit adults living in the Yorkshire and Humber region, so from the options available on Facebook we selected adults (aged over 18 years) living in this region. This was people living in the counties of North Lincolnshire England, North-East Lincolnshire England, West Yorkshire England, South Yorkshire England, North Yorkshire England and East Riding of Yorkshire England. The second boost aimed to attract more men to respond, to do this we changed the picture in the post to be more attractive to men. The picture on this boost was of a plate of beef and Yorkshire pudding with vegetables, a typical dish of the region. The third boost aimed to attract younger adults, a different picture was used again, this time of a healthier plate of salad foods with some colourful and novel ingredients.

We also offered an optional Prize Draw in the Facebook posts used to recruit adults. The prize(s) on offer were 1 × £100, 4 × £50 and 10 × £10 shopping vouchers. The Prize Draw was accessed by an URL link from the main survey to a separate questionnaire. Email/phone contact details and name were requested so that winners could be notified, but Prize Draw contact information was held separately from the main survey, which was anonymous. Once the random draw was made and prizes had been sent out, all contact details were deleted. 

At the end of the main questionnaire and at the end of the prize draw questionnaire a URL link to myfood24^®^ was provided so that participants could proceed to complete an anonymous diet diary for the previous day. Once the study had closed to recruitment the Facebook page was edited, the call to action button was changed to ‘Follow’ and all URL links were removed.

### 2.2. Ethics

Due to the tight timeframe for the study we used a proportionate ethical review process. This is used for research where the ethical issues raised are minimal. This meant that participants were not asked for their name or contact details. This had implications for the study delivery, including linkage between online survey and dietary assessment software and participants were not able to withdraw from the study once they had submitted their responses. 

### 2.3. Data Collected

The main questionnaire for the One Day: Diet in Yorkshire and Humber survey was set up in Online Surveys. Local public health colleagues provided input to the questionnaire. There were 17 questions. Two questions assessed eligibility, and one question related to consent. The remaining questions covered basic demographic information. Two questions related to food insecurity experience. The final question asked for a memorable word which was to be repeated in the next part of the survey. This was needed because General Data Protection Regulation (GDPR), a set of EU rules on data protection and privacy no longer allow for tokens to be passed between survey tools allowing them to be linked. 

The 24 h dietary recall for the One Day: Diet in Yorkshire and Humber survey was conducted using the research version of the nutritional analysis software myfood24^®^ [18]. Participants were asked to recall the foods and drinks they had consumed over 24 h the previous day from midnight to midnight (yesterday). They were not required to record the exact time of food intake. myfood24^®^ uses meal events (breakfast, lunch, evening dinner, snack, drink) to prompt the input of information.

The 24 h dietary recalls were submitted anonymously, but participants entered the memorable word used in the main Questionnaire in the further comments/feedback to link the two records. As the 24 h dietary recall was anonymous, participants could not start a food diary and return to complete it later; the food diary was reset after 24 h. 

myfood24^®^ was set up with four UK food composition databases McCance and Widdowson 7 (generic), McCance and Widdowson 7 (branded), dietary supplements and South Asian dishes. On submission/completion of each 24 h dietary recall, myfood24^®^ displayed a downloadable nutrient summary to the user which compared the main nutrients (Energy, Protein, Fat, Saturated fat, Carbohydrate, Total sugars, Salt, Fibre) in the food and drink intake reported for that day with UK Adult reference intake values. 

### 2.4. Data Analysis

Data from the short online questionnaire (including consent) from Online Surveys and food and drink intake from myfood24^®^ was downloaded and stored according to University of Leeds Research Data Management guidelines. Data was stored as Excel or STATA files in University of Leeds approved cloud computing software (OneDrive) using password-protected files which cannot be accessed by the public, students, or researchers other than those working on this project.

Once the Main questionnaire was closed, dropout numbers at each stage of the questionnaire were ascertained. Results data from submitted questionnaires were saved in Excel and a summary report was created in Online Surveys. Online Surveys automatically generated a unique 18 digit response number and a date and time stamp for each submitted questionnaire.

Data was checked in Excel and STATA for missing observations, possible duplication of responders, and plausibility, for example, in relation to self-reported height and weight and to determine whether any participants or incomplete questionnaires should be excluded. New variables for Body Mass Index (kg/m^2^) and BMI classification were generated in STATA, based on participants’ reported heights and weights after the removal of implausible outliers.

A sample size of 2000 participants was selected linked to the ability of the study to detect the proportion of people at risk of food insecurity. Using regional data suggesting that around 4% may be at risk [19]. A one sample proportion test was used estimating 6% to be food insecure to be detected, with a 95% significance level and power of 80%. This generated a total sample size of 853. Allowing for drop out and some stratification of data a sample of 2000 was the aim. Summary statistics for each variable were run in STATA 17.0 MP and compared with the summary report generated in Online Surveys. Participants BMI and reported food insecurity were examined by location (local authority), sex, age, occupation and whether adult respondent lives in a household where there are children. BMI was examined by reported food insecurity. Descriptive statistics for intakes of the main nutrients: energy, protein, fat, saturated fat, carbohydrate, total sugars, fibre, sodium, alcohol, iron and intake of fruit and vegetables and water were generated for those who had completed myfood24^®^. 

The memorable word and the time and date stamp of each submitted main questionnaire were compared with memorable word and the time and date stamp of each submitted 24 h dietary recall in myfood24^®^, to ascertain which records could be reliably matched (submitted by the same individual). This was done initially in Excel and then by applying a “fuzzy match” technique in a Jupyter notebook using Python.

## 3. Results

### 3.1. Engagement with Facebook Boosted Posts

The first boosted post was targeted to reach all adults living in Yorkshire and Humberside. The initial boost reached 19,277 people in the first few days and generated 208 link clicks to the main questionnaire, but the boost was then halted by Facebook. Once the issue was resolved the same post was boosted again for 10 days and was seen at least once by over 62.3 k individuals, of whom 73% were women and 27% men. The majority reached were aged 45 y and over. Post engagements (actions that people take involving the boosted post) were 2522 or 4.1% of the audience reached. Of these, there were an estimated 1377 link clicks and 601 were post reactions. Average cost per engagement (total amount spent divided by post engagements) was £0.20, while average cost per link click to the main questionnaire was £0.36.

The second boost, which ran for 9 days, aimed to reach more men in the region. An estimated 13,736 men saw the post at least once. Post engagements were 1330 (9.7% of the audience reached) and the average cost per engagement with the post was £0.15. There were 44 link clicks to the main questionnaire, costing £4.51 on average.

The third boost, aiming to reach younger adults 18 y to 54 y, ran simultaneously for the same 9 days. This reached 22,202 people. No one aged 55 y or over was reached, though the majority (82%) reached were women. There were 2316 post engagements (10.4% of those reached), the majority of which were reactions, not link clicks. The average cost per engagement was £0.08. There were 163 link clicks, costing £1.19 on average.

Overall, just under £1000 (£893) was spent on 3 boosts. The Facebook page was set up to allow participants to leave comments. The boosted posts received 9, 82, 14 and 6 comments, respectively and included some useful feedback. 

### 3.2. Sample Size

An estimated 1.8 k link clicks were reported by Facebook from the four boosts. 638 potential recruits left at the eligibility question relating to living in the region. 74 left at the consent question. A full count of numbers leaving at each stage of the main questionnaire can be found in Appendix B. 1430 participants completed the main questionnaire, although 2 were later excluded. This was 58% of those who started it (2475) and 92% of those who reported that they were eligible and gave consent (1559). The price for recruiting each person through Facebook who completed the main questionnaire was 63 p. 1090 participants entered the prize draw and 673 participants completed the diet diary, which was 51% of those who started the diary (1317). 

### 3.3. Data Checking

Assessment of the memorable word approach which aimed to link the online survey with the diet diary found that 60 words appeared more than once. This suggested that some individuals may have submitted more than one questionnaire. Duplicate submissions of anonymous responses are always a possibility, but usually cannot be identified. Responses for ten duplicate/highly similar memorable words were compared. The comparison showed that while some responses with the same memorable word had identical sets of answers (likely to be the same individual), others had non-matching postcodes and different answers (likely to be responses from different individuals). The decision was made to keep all responses; it was uncertain exactly which responses were duplicates and the overall number of potential duplicate responses was small (<2%). 

With few exceptions partial postcodes given for the home address corresponded correctly with the local authority area. A maximum of four characters could be entered for the partial postcode (as some partial postcodes are 4 characters); most anomalies could be explained by the participant entering the first four characters of the home postcode rather than their shorter two or three character partial postcode E.g. LS64 (not in Leeds) instead of LS6 (in Leeds). 

### 3.4. Participant Characteristics

The largest number of responses were from North Yorkshire (280), followed by Leeds (236). 89% of responders were women. 80% of responders were over 45 years old. The largest number of responses were from 56 to 65 years age group (439), followed by 66 to 75 year age group (346), together making up more than half the sample. 97% of responders stated they were of white ethnicity. The largest number of responses were from people who described themselves as retired (563) making up 40% of respondents. The next largest group had managerial and professional occupations (398) making up 28% of respondents. Unemployed respondents (49) made up 3.5% of the sample. Only 16% of responses were from people who had children (under 18 years of age) living in their household.

The mean self-reported heights and weights (n = 1012) resulted in a BMI of 27.9 kg/m^2^ (SD 6.5). Of the sample reporting height and weight, 38% were healthy weight (18.5 to 24.9 kg/m^2^), 33% were overweight (25 to 29.9 kg/m^2^) and 28% obese (30 kg/m^2^ or above). These values are in line with those reported by the Health Survey for England 2019 [20] which estimates that 28.0% of adults in England were obese and a further 36.2% were overweight but not obese. 

The highest percentage of overweight and obesity was observed in Hull local authority (80%), though numbers were small, Barnsley (77%) and Doncaster (76%). Highest percentage of healthy weight was observed in York (78%, with small numbers), Sheffield (51%) and North Yorkshire (41%). 

### 3.5. Food Insecurity

In total 22% of respondents reported experiencing moderate food insecurity (worried about obtaining food or reduced the quality and quantity of food) during 2021 (Table 1). Less than 4% experienced moderate food insecurity often, but 18% experienced moderate food insecurity sometimes. 9% of respondents reported severe food insecurity (skipped meals or experienced hunger) during 2021. Only 1% experienced severe food insecurity often, but almost 8% experienced severe food insecurity sometimes. 

Both moderate and severe food insecurity were reported most often in younger age groups. In relation to BMI, in the moderate food insecurity group, those who responded often had the highest BMI 34.8 kg/m^2^ compared to those answering never (BMI 27.4 kg/m^2^). This was similar for the severe food insecurity group with the highest mean BMI (33.4 kg/m^2^) for those responding often. Moderate food insecurity was reported most often by people who were unemployed (65% of them). Small employers/own account workers and people working in supervisory/technical roles (perhaps with less steady or smaller incomes) also reported experiencing moderate food insecurity more than other occupations. Some people preferred not to say their occupation, of whom 36% experienced moderate food insecurity. Severe food insecurity was reported most often by people who were unemployed (10% often and 41% sometimes). A greater proportion of participants from households with children reported experiencing moderate (28%) and severe (14%) food insecurity than participants from households without children.

### 3.6. Linking Records

Examination of the memorable words in the 1428 submitted main questionnaires (Online Surveys) and in the 673 completed/submitted 24 h dietary recalls (myfood24^®^) showed that only 51 participants had entered their memorable word in both records. Assuming that participants may complete the main questionnaire and then go on to complete the food diary shortly afterwards led to examination of the date and time stamps of both systems. There was a considerable overlap of records as the recruitment campaign gained momentum. Using Python in a Jupyter notebook, the date and time stamps for both records were sorted into chronological order. The information was then examined to see where records might correlate. The records already paired with matching memorable words clearly showed that adjacent rows were not necessarily a pairing. The time taken between finishing the main questionnaire and starting the dietary recall varied considerably between individuals, such that date and time stamps were not a reliable method of linking the data. 

### 3.7. Food Diary Results

667 dietary recalls containing some data were recorded in myfood24^®^. On average, participants added 13 to 14 food/drink entries per diary. 19% of entries were at breakfast, 23% at lunch, 29% at dinner, 12% were snacks and 17% in the drink category. Approximately 95% of dietary recalls were reported to be representative of a “typical” day and an estimated 40% of dietary recalls included vitamins, minerals, or other supplements (such as multivitamins, Vitamin D, calcium, or cod liver oil). Of all food reported, the food categories reported most often were: hot drinks (10%), vegetables and potatoes (10%), dairy and eggs (7%). The food groups bread, milk and dairy drinks contributed 5% each with meat and poultry category at 4%.

Overall nutrient and food intakes reported from myfood24^®^ are shown in Table 2. Mean energy intakes reported were slightly lower than the average from the UK National Diet and Nutrition Survey. However, this is likely due to a higher proportion of older women in our sample. The macronutrient values also reflect this difference. Reported fat and carbohydrate value were lower than reference intakes. Fibre intake was particularly low. Fruit and vegetable intakes represented 4 portions consumed per day. 69% of dietary recalls did not report any intake of alcohol. 

Measures of sustainability metrics from the dietary intakes were also calculated, for the first time, from myfood24^®^. The mean greenhouse gas emissions were 5.9 (kg CO_2_ eq/day) (95% CI 5.6, 6.1); land use 9.0 (m^2^ year/day) (95% CI 7.6, 10.3) and water use 582 (L/day) (95% CI 551, 611).

## 4. Discussion

This study has demonstrated the potential to use social media, in this case Facebook, to recruit a large sample in a short timeframe, one month. Our initial plan was to recruit up to 2000 adults from the region. Around 2.5 k adults started the questionnaire with 1.5 k completing it. Longer recruitment time or higher budget for the boosts would have led to larger numbers if required. Online survey tools provided a quick way to obtain key nutritional information. Facebook advertising offers an easy, rapid, and economical means to recruit a study sample. Others have achieved partially representative samples of middle-aged and older adults for health survey research [1]. Our sample was older, and the majority of respondents were women, despite efforts to target younger adults and men. This may have been due to the topic of the survey or reflect Facebook user demographics. As Facebook uses a non-random targeting algorithm, caution needs to be used in applications for certain types of research. However, it would have been possible to weight our sample results to be more representative of the regional population distribution.

Use of social media enables rapid recruitment of large samples [4]. Declining landline use makes other recruitment methods such as random digit telephone dialing less effective, 22% of UK households no longer have a landline and this number is increasing [21].

One particular frustration in this study was the lack of direct linkage between the online survey system used and the detailed nutritional analysis software. Our attempt to keep things anonymous (due to time constraints creating the need for a rapid ethics review) prevented us from linking the two sets of data in a reliable way. In the past, this online survey system has allowed passing of a unique link to other systems. However, due to the introduction of GDPR this was stopped to minimise the potential to identify individuals. Both systems used to collect data are ISO27001 certified, this is an international information security management standard that focuses on confidentiality, integrity and availability of information. Since this survey, myfood24^®^ has been further developed to include standalone consent questions, and a self-generated unique ID which could be copied and entered into another online tool if needed. More time would also have allowed for the more intensive and time consuming ethics procedures required to collect personally identifiable information to link the surveys.

Reported levels of food insecurity were relatively high in this sample. Our analysis showed higher values than the recent Food and You survey, which also uses an online survey methodology, ‘push-to-web’. In the Food and You survey, 15% of respondents were classified as food insecure [22]. Household food insecurity levels have increased by 60% since the first six months of the pandemic. Particularly concerning for public health is that nearly half of households on Universal Credit have experienced food insecurity in the past six months, making this group particularly vulnerable [23]. There continue to be significant regional inequalities in food insecurity across the UK, with Yorkshire and the Humber being at the higher end, with 15.1% reporting food insecurity in the last month. London had the lowest rate at 11.1% [23], suggesting there is a way to go on the levelling up agenda. 

In addition to cross-sectional surveys, social media could also be used to deliver interactive interventions to a wide population, including the most vulnerable groups. Recent systematic reviews of app-based mobile interventions suggested they could improve nutrition behaviours and nutrition related health outcomes [24,25]. A further systematic review comparing studies assessing interactive social media interventions with non-interactive social media interventions also found a positive effect on increasing physical activity and well-being. However, the 8 studies included exploring diet quality showed variable effect, with no overall improvement [2].

This survey has demonstrated the potential to collect detailed dietary data on a large sample of adults quickly and cheaply. Diet tracking apps have been scored well in terms of usability [26]. myfood24^®^ usability testing indicates that it is suitable for use with both adults and adolescents [27]. Respondents to our main questionnaire were 89% female and 60% were aged over 55 years, these older women are likely to have lower food intakes than younger women or men. Results from myfood24^®^, show mean energy intakes are lower than reference values, but our results compare well with the NDNS, national survey data [11]. The Reference Nutrient Intake (RNI) for protein for adults is 0.75 g protein per kg body weight. So our intakes suggest they would be enough for someone who is 80 kg (or 12 stone 8 lbs). Potentially of concern here are the low fibre and iron intakes compared to reference values. The NDNS data is now also using an online 24 h dietary recall tool from Year 12 (2019 to 2020) to collect dietary data from participants. This has a relatively low response rate (30%) from those invited to take part. 

There were limitations to this study, not least the challenges presented by connecting between systems. Only Facebook was used as the social media service being tested, this is the largest active social media platform [5]. We may have found different results if other systems such as Instagram or Twitter had been used. In addition, the response was skewed towards older females completing the survey. Traditional survey methods also have non-representative samples taking part. No interviewers were used who could potentially have probed for more detail. However, nutrient intakes were similar to interviewer checked NDNS data suggesting reliable reporting. The online approach used does have potential. The survey costs were much lower than an interviewer administered survey would have been. A large sample was obtained within three weeks of running the survey. The geographical area was targeted well through Facebook. Users were able to complete the online survey and myfood24^®^ and analysis was quick with no additional coding required.

Future surveys using this approach could be carried out regularly, and use unique identification information allowing linkage between tools, to monitor population intakes and evaluate public health interventions supporting improved eating behaviours.

## 5. Conclusions

This survey has shown that by using a FB boost survey approach it is possible to obtain regionally focussed data on a large sample, quickly and cheaply. Responses were obtained from people across the whole region. However, the majority of responses were from older women using this social media platform. More younger adults could potentially have been recruited if other social media sites had been used [5]. The cost per individual recruited was less than £1.00 to FB. This is considerably cheaper than most researcher led recruitment strategies or paid for marketing organisations. However, the ability to link data from individual digital tools needs to be streamlined.

Detailed dietary information was collected for the first time in this region of the UK, showing similar nutrient intakes to the national picture, but with worrying levels of reported food insecurity. This method of recruitment and data obtained is relevant to local and national policy makers. This approach could be used to track progress towards dietary guidelines or assess the impact of food reformulation. These tools can be used to monitor and evaluate public health programmes and interventions in relation to improving food and nutrient intakes.

## Figures and Tables

**Table 1 nutrients-14-05322-t001:** Responses to food insecurity questions.

Age, Years	18 to 25 (n 44)	26 to 35 (n 97)	36 to 45 (n 146)	46 to 55 (n 287)	56 to 65 (n 439)	66 to 75 (n 345)	76 to 85 (n 57)	86+ (n 7)	Total
Moderate (n 1426):									
% Never	43%	64%	73%	74%	83%	85%	79%	100%	78%
% Often	9%	7%	8%	5%	3%	1%	2%	0%	4%
% Sometimes	48%	29%	20%	21%	15%	14%	19%	0%	18%
Severe (n 1425):									
% Never	70%	87%	86%	87%	94%	95%	93%	100%	91%
% Often	5%	4%	3%	3%	0%	0%	0%	0%	1%
% Sometimes	25%	9%	12%	10%	6%	4%	7%	0%	8%

**Table 2 nutrients-14-05322-t002:** Food and nutrient intakes reported.

Mean Intakes/Day	One Day: Diet Study	NDNS *	Reference Intake *
N adults	667	<500 (19 to 64 years)	-
Energy kJ	6281 (SD 2733)	7690	8400 kJ women 10,500 kJ men
Energy kcal	1497 (SD 653)	1828	2000 kcal women 2500 kcal men
Protein g	61 (SD 26)	76	50 g
Total fat g	59 (SD 32)	70 36% total energy	<70 g or no more than 33% total energy
Saturated fat g	21 (SD 14)	25 12% total energy	<20 g or no more than 10% total energy
Carbohydrate g	168 (SD 85)	219	260 g
Total sugars g	65 (SD 45)	n/a	90 g
Free sugars g	n/a	50 10% total energy	<30 g or no more than 5% total energy
AOAC Fibre g	18 (SD 11)	19.7	30 g
Sodium g	1.8 (SD 2.2)	n/a	Less than 2.4 g
Iron mg	9.6 (SD 5.0)	10.5 (Women 9.4 mg)	8.7 mg men >18 y. 14.mg women 19–50 y. 8.7 mg women >50 y.
Total Fruit and Veg. g/portions	322 g (SD 277) or 4 portions	4.3 portions	5 a day 5 × 80 g portions = 400 g
Alcohol g/units	8 8 g (SD 20) 1.1 unit	10.8	No more than 14 units (112 g) of alcohol/week

* National Diet and Nutrition Survey Rolling Programme Years 9 to 11 (2016/17–2018/19); Daily Reference Intakes or UK Eatwell guidance.

## Data Availability

Data supporting this paper can be obtained from the corresponding author on request.

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
