# Peer review of "Using Social Media to Collect Dietary Information for Public Health Policy"

_nutrients, 2022, doi:10.3390/nu14245322_

Round 1

Reviewer 1 Report

General comments, it is an interesting manuscript that propose the use Web-based recruitment through social media to track the local population food intakes. Pioneering approach that can be replicated.

I would like to presented minor comments, as shown below.

Introduction

Line 55 – edit the letter type; 

Material and methods

line 60 – reference the police criteria and indicators used to build the questionnaire - analyzed constructs should be mentioned;

Authors's should indicate the data collection procedures and provide a sample of the questionnaire used;

Line 89, 122, 129 – revise the phrase due to word repetitions;

The generation of engagement with Facebook boosted posts could be better explained on method section, as well as sample size definition criteria and data checking procedures.

Author Response

General comments, it is an interesting manuscript that propose the use Web-based recruitment through social media to track the local population food intakes. Pioneering approach that can be replicated.

  • Thank you.

I would like to presented minor comments, as shown below.

Introduction

Line 55 – edit the letter type; 

  • Changed to normal type, I had wanted to use italics to suggest the importance of the sentence. However, if that is not allowed I have removed it. If it is allowed, I would like to retain italics here.

Material and methods

line 60 – reference the police criteria and indicators used to build the questionnaire - analyzed constructs should be mentioned;

  • Formal construct analysis was not planned. The aim of the project was to test the new approach to participant recruitment rather than any novel analysis.

Authors's should indicate the data collection procedures and provide a sample of the questionnaire used;

  • Detail on the questionnaire content is provided further down the methods section 2.3 Data Collected. This has been added to line 69 for information.
  • The myfood24 tool is a standard format. A link to the website has been added (ref 13) so that readers can see the tool in more detail if needed.

Line 89, 122, 129 – revise the phrase due to word repetitions;

  • Edits and removals have been made to these sentences to avoid repetition.

The generation of engagement with Facebook boosted posts could be better explained on method section, as well as sample size definition criteria and data checking procedures.

  • More detail on the different Facebook boosted posts has been added (lines 103-106).
  • A sample size calculation has been provided (lines 172-176)
  • More detail has been added on data checking in the methods (line 168)

Reviewer 2 Report

Introduction

This part needs to be improved. Hardly any studies are cited to argue why this research is important.

The following questions should be answered:

- Why do social networks have to be the focus of study to find out if they are a source of information on healthy eating?

- Why Facebook and not, for example, Instagram? Recent studies show that Instagram is the most popular social network for information on food.

- Why this audience and in this area?

- What other studies address this object of study and why is it important to study it, what does this study contribute?

What are the objectives of the research?

MATERIALS AND METHODS

From the reading we understand that this is an exploratory study, correct?

Have any statistics been used to cross-check the data?

CONCLUSIONS

The conclusions are insufficient. We should begin with a brief summary of the results presented, try to compare these results with previous research (hence the need to improve the introductory section) and, finally, consider what this work contributes to public policies on the creation of healthy eating habits among the population.

Author Response

Introduction

This part needs to be improved. Hardly any studies are cited to argue why this research is important.

  • Further references and justification have been added lines 49-54.

The following questions should be answered:

- Why do social networks have to be the focus of study to find out if they are a source of information on healthy eating?

  • Of course, social networks do not have to be a focus. However, they do provide access to a wide spectrum of the population including those who might not engage in research through other approaches. I have added a new sentence (lines 38-9) to explain this a bit further.

- Why Facebook and not, for example, Instagram? Recent studies show that Instagram is the most popular social network for information on food.

  • We wanted to explore one route in the first instance, Facebook has more users in the UK than Instagram. Of course there are many others and this approach to recruitment will grow. A sentence to explore this comment has been added to the discussion (lines 382-384)

- Why this audience and in this area?

  • We wanted to explore food and nutrient intakes in adults living in our local region for use with policy makers (one of whom is a co-author).

- What other studies address this object of study and why is it important to study it, what does this study contribute?

  • There are no other studies which have explored recruiting participants using social media for policy maker tracking of food and nutrient intakes in their local population.
  • As mentioned above more references have been added in the introduction to justify the importance of this work to policy makers.

What are the objectives of the research?

  • The aim has been re-written to include objectives (lines 61-63)

MATERIALS AND METHODS

From the reading we understand that this is an exploratory study, correct?

  • Yes

Have any statistics been used to cross-check the data?

  • I am not sure what is being asked here? We were unable to cross-check between online questionnaire and dietary intake. This was a frustration as expressed in the write up discussion lines 336-338.

CONCLUSIONS

The conclusions are insufficient. We should begin with a brief summary of the results presented, try to compare these results with previous research (hence the need to improve the introductory section) and, finally, consider what this work contributes to public policies on the creation of healthy eating habits among the population.

  • Further detail has been added to the conclusion to hopefully better express these aspects.

Round 2

Reviewer 2 Report

he introduction is still insufficient. A noticeable improvement was called for and only three more references are provided. Although the topic is new, I insist, there is a remarkable literature on the subject that is not being referenced. Furthermore, the objectives, research question and, if necessary, the hypothesis are still missing.
With this starting point, the conclusions (only two lines have been added) are still insufficient. In the revision, a modification of this section was proposed based on the following parameters:
- Start with a brief summary of the results.
- Compare the results with similar research. To this end, it is necessary to make and improve the introduction
- To deepen the approach to the conclusions: what does this research contribute?

Translated with www.DeepL.com/Translator (free version)

Author Response

he introduction is still insufficient. A noticeable improvement was called for and only three more references are provided. Although the topic is new, I insist, there is a remarkable literature on the subject that is not being referenced. Furthermore, the objectives, research question and, if necessary, the hypothesis are still missing.

  • the introduction is now doubled in length with more references to use of social media in nutrition policy and study recruitment added
  • the objectives have been numbered 1 to 3 and the aim further detailed. We were not testing a specific hypothesis in this study, so none has been added.

With this starting point, the conclusions (only two lines have been added) are still insufficient. In the revision, a modification of this section was proposed based on the following parameters:
- Start with a brief summary of the results.
- Compare the results with similar research. To this end, it is necessary to make and improve the introduction
- To deepen the approach to the conclusions: what does this research contribute?

  • the conclusion is now over twice as long.
  • I have aimed to follow the suggestions above as an approach.